# Development of a bioassay method to test activity of cry insecticidal proteins against *Diatraea* spp. (Lepidoptera: Crambidae) sugarcane stem borers

Juan Sebastián Ángel-Salazar[1], Claudia Echeverri-Rubiano[1], Jairo Rodríguez-Chalarca[2], Jershon López-Gerena[1], Rafael Ferreira dos Santos[3], Juan Luis Jurat-Fuentes[3], Alexandra M. Revynthi[4], Germán Vargas[4]*

1 Colombian Sugarcane Research Center (Cenicaña), Florida, Colombia, 2 Alliance Bioversity-CIAT (The International Center for Tropical Agriculture), Palmira, Colombia, 3 Department of Entomology and Plant Pathology, University of Tennessee, Knoxville, Tennessee, United States of America, 4 Department of Entomology and Nematology, Tropical Research and Education Center, University of Florida, Homestead, FL, United States of America

* entomogav@gmail.com

**Citation:** Ángel-Salazar JS, Echeverri-Rubiano C, Rodríguez-Chalarca J, López-Gerena J, dos Santos RF, Jurat-Fuentes JL, et al. (2023) Development of a bioassay method to test activity of cry insecticidal proteins against *Diatraea* spp. (Lepidoptera: Crambidae) sugarcane stem borers. PLoS ONE 18(10): e0292992. https://doi.org/10.1371/journal.pone.0292992

## Abstract

The genus *Diatraea* (Lepidoptera: Crambidae) includes stem borers representing the most critical sugarcane pests in the Americas. Colombia's most widely distributed and damaging *Diatraea* species include *Diatraea saccharalis*, *D. indigenella*, *D. busckella*, and *D. tabernella*. The reduced efficacy of biological tools commonly used in controlling several species highlights the importance of evaluating alternative management strategies, such as transgenic plants expressing insecticidal proteins from the bacterium *Bacillus thuringiensis* (Bt). The selection of optimal Bt insecticidal proteins for *Diatraea* control depends on bioassays with purified Bt proteins. Because there is no described artificial diet for borer species other than *D. saccharalis* and availability of most purified Bt toxins is restricted, this study aimed at developing a bioassay method using fresh corn tissue and providing proof of concept by testing susceptibility to the Cry1Ac insecticidal protein from Bt. Toxicity was evaluated with a single Cry1Ac dose applied directly to corn discs. Stem borer mortality after seven days was higher than 90% for all four tested *Diatraea* species, while control mortality was below 8%. In addition, we observed that Cry1Ac caused more than 90% weight inhibition in all survivors and delayed development. These results validate the use of this method to determine mortality and growth inhibition due to the consumption of the Cry1Ac protein in each of the *Diatraea* species. Furthermore, this method could be used to assess other entomopathogenic substances to control these insect pests.

## Introduction

Sugarcane is an important crop in the tropics and subtropics worldwide, where it is mainly grown to produce sucrose, alcohol, electricity, and panela (Jaggery or non-centrifuged sugar)

**Data Availability Statement:** All relevant data are within the paper and its Supporting Information files.

**Funding:** The authors thank Alliance Bioversity-CIAT (The International Center for Tropical Agriculture) for providing the logistical support required to conduct this study and the Colombian Sugarcane Research Center (Cenicaña) for technical and financial support to the first author. The NC246 Multistate Hatch project from the USDA National Institute for Food and Agriculture (NIFA) and the University of Tennessee provided support for insecticidal protein production. The funders had no role in study design, data collection and analysis, decision to publish, or preparation of the manuscript.

**Competing interests:** The authors have declared that no competing interests exist.

[1, 2]. Colombia has one of the highest global sugarcane productions per area unit, yet stem borers cause significant yield losses, ranging from 100 to 143 Kg of sucrose per hectare, and per each percent of bored internodes [3, 4]. Among the lepidopteran species described as sugarcane stem borers [3], most belong to the family Crambidae [5] and to the *Diatraea* genus in the Western Hemisphere. The most updated report on the *Diatraea* genus indicates that 41 species are present in the Americas [6, 7].

*Diatraea* stem borers attack sugarcane in all its developmental stages, with production losses depending on the crop stage and damage caused [8]. Attacks on young plants can significantly damage the leaves and stem tissues and cause the syndrome called "dead heart" (apical bud death). However, this syndrome only reduces sugarcane production when most shoots are affected and if insect feeding continues for at least a month, resulting in reductions up to 30% of harvested sucrose [9]. In addition, bored holes facilitate infection by the fungi *Colletotrichum falcatum*, resulting in rotting and reduced sucrose content. Losses caused by *Diatraea* stem borers have been estimated as a reduction in both total biomass production (up to 109 Kg of sucrose per hectare and per each percent of bored internodes) and sucrose content (up to 34 Kg of sucrose per hectare and per each percent of bored internodes) [4, 5].

At least six species of the genus *Diatraea* have been reported to affect Colombia's sugarcane crops [7, 10]. Although some sugarcane industries rely on chemical control for the borer's management; in Colombia, these pests have been mainly managed using biological control agents attacking the egg and larval stages [11]. However, decreased effectiveness of several parasitoids has been observed on *Diatraea* species found in the Cauca River Valley (CRV), the most industrialized area for sugarcane production in Colombia, primarily due to the expression of host resistance from *Diatraea tabernella* Dyar and *Diatraea busckella* Dyar & Heinrich to the main parasitoids used in the CRV [12]. As a result, searching for new management strategies complementary or alternative to current control tools is needed.

One control option is the use of Cry [13, 14] and Vip3 [15] insecticidal proteins from the bacterium *Bacillus thuringiensis* (Berliner) (Bt). These proteins target midgut cells and disrupt the epithelial barrier, allowing passage of bacteria into the main body cavity resulting in lethal septicemia [16]. Bt proteins and insecticides' beneficial hallmarks are their high specificity and established history of safety to vertebrates and non-target insects [13]. Given the boring activity displayed by *Diatraea* spp. in sugarcane, transgenic plants producing Bt insecticidal proteins are the preferred alternative for controlling sugarcane stem borers [17]. The efficacy of these insecticidal proteins is commonly evaluated in bioassays under controlled conditions with the purified protein incorporated in an artificial diet and monitoring survival and development [18], or by direct observation of their efficacy in plants in the field [19]. Several studies have assessed the activity of diverse Bt proteins, such as Cry1Ab, Cry1Ac, Cry1F, Cry2Ab, and Vip3Aa20, on sugarcane stem borers, including *Diatraea saccharalis* (F.), *Diatraea flavipennella* Box and *Elasmopalpus lignosellus* (Zeller) (Lepidoptera: Pyralidae) [15, 17, 20–23]. These studies were facilitated by the availability of an artificial diet capable of sustaining *D. saccharalis* development and reproduction [24, 25].

In Colombia, four *Diatraea* species present the most significant distribution and economic impact on sugarcane: *D. busckella*, *D. indigenella*, *D. saccharalis* (F.), and *D. tabernella* [7, 11]. Except for *D. saccharalis*, no artificial diet is available for testing insecticidal protein activity on other species. Additional research is warranted in relation to the development of an artificial diet for the additional *Diatraea* species (i.e., *D. busckella*, *D. indigenella* and *D. tabernella*). Investing additional effort will yield appropriate diets for these species. However, a potential challenge arises from the necessity of utilizing distinct diets for each species, which could complicate the comparability of survival and biological responses in trials where comparisons among different species are required. Performing experiments using a species-specific diet,

would increase the variation in the set-up, hindering this was straightforward comparisons among *Diatraea* species. In solving this current knowledge gap, this study focused on establishing an alternative method using a fresh food substrate for evaluating susceptibility to Cry proteins and providing proof of concept by testing Cry1Ac susceptibility in *D. busckella*, *D. indigenella*, and *D. tabernella*. Results using corn discs as a food source support high susceptibility to Cry1Ac in all tested *Diatraea* spp. and validate this method for future trials using alternative Bt proteins, commercial pesticides, or plant tissues expressing insecticidal proteins.

## Materials and methods

### Insects

The entomology laboratory at the Colombian Sugarcane Research Center (known by the Spanish acronym Cenicaña), located in Florida, Valle del Cauca, Colombia, supplied the insects used in this study. All insect colonies were obtained from sugarcane collections in the CRV and maintained in a climate-controlled room at a temperature of $25 \pm 1°C$, $70 \pm 4\%$ RH, and 12:12 L:D photoperiod; on slices of tender corn cob previously disinfected by flame sterilization using laboratory burners for 15–20 seconds. All insects used in bioassays corresponded to first generation individuals from field-collected insects. Corn cobs for rearing larval feeding were sliced into pieces ($0.5 \pm 0.1$ cm thick) using an 11–1/4–inch Alligator Slicer (Alligator of Sweden, Upplands Vasby, Sweden) and subsequently cut into discs with a hole punch (1.4 cm diameter). The maize cultivar SV 1035 was grown conventionally (mineral fertilization with NPK and manual weed control) under field conditions, but without using insecticides, while ensuring it was situated 2 kilometers away from other commercial corn crops. This strategic separation aimed to prevent any potential cross-contamination. The harvesting process was done manually at 70 days of the crop cycle. Coincidentally, this same source of corn was also employed as a substrate to sustain the various *Diatraea* species stock colonies at the Cenicaña entomology laboratory. To confirm the absence of any expression of Bt proteins, we conducted regular assessments of survival within the stock colonies. Corn was used instead of sugarcane because it allows the development of all borer species and presents a lower level of fermentation [26].

### Insecticidal protein purification

The Cry1Ac protoxin was produced in cultures of *Bacillus thuringiensis* strain HD73, obtained from the *Bacillus* Genetic Stock Center (Columbus, OH, USA) in 1/3 tryptic soy broth (TSB) medium at 28°C with shaking for 72h. After collecting spores and Cry1Ac protein crystals by centrifugation at 6,000rpm for 30min, pellets were washed in 1M NaCl plus 0.1% Triton-X-100, and this collection and washing cycle was repeated twice until the final pellet was washed with MilliQ water. The Cry1Ac crystals were solubilized at 30°C in carbonate buffer (50mM $Na_2CO_3$, 0.1M NaCl, 0.1% 2-mercaptoethanol, pH10.5), and the solubilized Cry1Ac protoxin activated by digestion with bovine TPCK-treated trypsin (Sigma, $\geq$10,000 BAEE units/mg) at a final concentration of 0.2mg/ml for 1 hour. Activated Cry1Ac toxin was purified by anion exchange in a Hitrap Q HP column (GE Healthcare) connected to an ÄKTA chromatography system (GE LifeSciences), using a linear gradient of 1M NaCl for elution. The purified Cry1Ac toxin was quantified using BSA as a standard [27] and shipped frozen to Cenicaña for testing.

### Bioassay

Tests were performed on four *Diatraea* species, *Diatraea saccharalis*, *D. indigenella*, *D. tabernella*, and *D. busckella*, by exposing neonates (< 24 h after hatching) to cob corn discs

prepared as described before, 70–75 days after planting, when kernels were at the milk stage (R3), using same cultivar and growing conditions described above. In general, the corn kernel at the milk stage is on the cusp of maturity, yet it remains soft and brimming with a milky fluid. This fluid is rich in carbohydrates, mainly in the form of sugars. At the start of the R3 stage, approximately 80% of its content comprises water. Non-peeled cob corns were store at 8˚C for no more than one week.

Bioassays were performed using 128-well bioassay trays (CD International Pitman, NJ), with a 2% agar plug poured and solidified at the bottom of each well to retain moisture and turgidity of the corn material used as feeding substrate. A corn disc with a surface area of 1.54 cm$^2$ and 0.5 ± 0.1 cm thick was incorporated into each well. The Cry1Ac protein in water containing 0.1%Triton-X 100 (total volume) was added to the surface of the discs. To account for any effects of moisturizing the cob corn discs with water and detergent (Triton-X 100), control larvae were exposed to discs treated with water and detergent. A concentration of 619.5 μg/ml in a final volume of 60 μl per well was used over an area of 1.54 cm$^2$, representing 24.1 μg/cm$^2$ of Cry1Ac. This lethal diagnostic dose was selected in assessing toxicity based on previously reported LC$_{50}$ values for Cry1Ac from diet surface bioassays with *D. saccharalis* [28–30] and *D. flavipennella* [21], but also based on trial-and-error observations trying to guarantee coverage and efficacy of the application over the fresh tissue.

The volume of 60 μl was uniformly applied onto the corn tissue using a 100 μl Eppendorf pipette. The tray was then balanced after each application to facilitate protein dispersion and absorption within the tissue. The corn disc substrate was allowed to absorb the solution for 1 h, then one first instar larva was added per disc. Wells were then sealed with air-permeable lids (CD International Pitman, NJ) and trays maintained at laboratory conditions mentioned previously. Mortality was scored 7 days after infestation, by assessing any level of activity in treated larvae. Larvae that survived the protein treatment were subsequently transferred to a control diet (corn cob discs treated with water and detergent) to observe further development.

Considering insect availability in the colony, *Diatraea* species were tested in separate weeks against their control. Each combination of *Diatraea* species and the protein were evaluated separately using a tray containing 16 wells. Simultaneously, a separate tray was designated as the control, at it contained water and detergent. As an air-permeable lid separated each group of 16 wells, it was considered an independent replicate, and the 16 wells were considered subsamples. Hence each combination of species and treatment had eight replicates, for a total of 128 larvae evaluated per species and treatment.

## Study variables

Variables were recorded 7 days after infestation, including larval mortality, weight, and instar. Percentage mortality (%M) was estimated from the number of dead individuals considering the number of individuals established in the bioassay using the formula %M = ($m$ / $N$) * 100, where $m$ is the total number of dead insects in the treatment, and $N$ is the total number of insects established in the treatment. Surviving larvae were weighted using a Mettler Toledo AB204 (Switzerland) analytical balance (precision ± 0.1 mg). These surviving individuals were then transferred to untreated corn to determine if they presented late mortality or if they managed to complete development.

The Growth Index for Survival (GIS) and Relative Growth Index (RGI) [31] were estimated considering the developmental stage of live and dead individuals. Instar information was gathered based on morphological characteristics for all the borer species under study [26]. Under the laboratory conditions described above, larvae of the four borer species tested took 3 days to grow from the first larval instar (L1) to the second instar (L2) and 4 days to grow from L2 to

the third instar (L3), totaling 7 days from L1 to L3. The GIS was estimated using the formula:

$$\text{GIS} = [(n_1 \times \text{L1}) + (n_2 \times \text{L2}) + \bullet\bullet\bullet + (n_x \times \text{L}_x)]/(\text{N} \times \text{L}_x);$$

Where $n_1$ was the number of individuals in instar L1, $n_2$ was the number of individuals in instar L2, and so forth, N corresponds to the total number of insects established in the treatment, and Lx was the maximum instar reached in the control treatment. After estimating the GIS for the control and treatment, the RGI for each species was calculated as the division between the GIS of the treatment and that of the control, where RGI < 1 indicates that the individuals survived or died in the early stages, RGI = 0 no development (or all killed in the first stage), and RGI = 1 complete development of individuals in the test group (or no difference from the control group) [32].

Percentage Growth Inhibition for Weight (%GI) [33] was calculated using the average larval weight from each replicate of the Cry1Ac treatments relative to the average weight in respective control treatments, calculated as follows:

$$\%\text{GI} = 1 - (\text{TWIT}/\text{TNIT})/(\text{TWIC}/\text{TNIC}) \text{ x } 100;$$

Where TWIT is the total weight of insects in the treatment, TNIT is the total number of insects in the treatment, TWIC is the total weight of insects in the control treatment, and TNIC is the total number of insects in the control treatment [33].

Contamination is not an issue in bioassays when using artificial diets as feeding substrates because they typically contain antimicrobial compounds. Our bioassays used fresh corn tissue (tender corn) as substrate, which is susceptible to fungal contamination. Therefore, we monitored and quantified fungal contamination levels. The percentage of corn contamination (%C) in each treatment was evaluated based on the presence of fungal mycelia on the corn surface as follows:

$$\%\text{C} = (\text{c}/\text{N}) * 100;$$

where c was the number of wells contaminated in the treatment, and N corresponds to the total number of wells established for that treatment.

## Statistical analyses

Experiments were arranged under a completely randomized design, with a factorial structure considering *Diatraea* species and treatments (Cry1Ac protein and control) as fixed effects and analyzed using generalized linear mixed models (PROC GLIMMIX, SAS 8.2) [34]. The 16 wells within each of the eight replicates were considered subsamples, and well effects were considered random. Each combination of *Diatraea* species and the protein was made performed once using a single borer generation. Mortality was analyzed as a proportion of dead larvae within each replicate, assuming binomial distribution, whereas indexes of the stage of development (GIS and RGI) and weight (%GI) were analyzed with available (alive) individuals, assuming Gaussian distribution. The mean weight of surviving larvae was also analyzed assuming Gaussian distribution. Fungi contamination of wells was estimated as the proportion of wells contaminated from the total 16 wells per replicate (subsample), and analyzed assuming a binomial distribution. Means were separated by Tukey tests ($\alpha$ = 0.05). In addition, linear regressions comparing the proportion of contaminated wells with the proportion of dead larvae in the controls and the proportion of contaminated wells with the proportion of larvae killed in the treatment were analyzed by linear regression (PROC REG, SAS 8.2) [34].

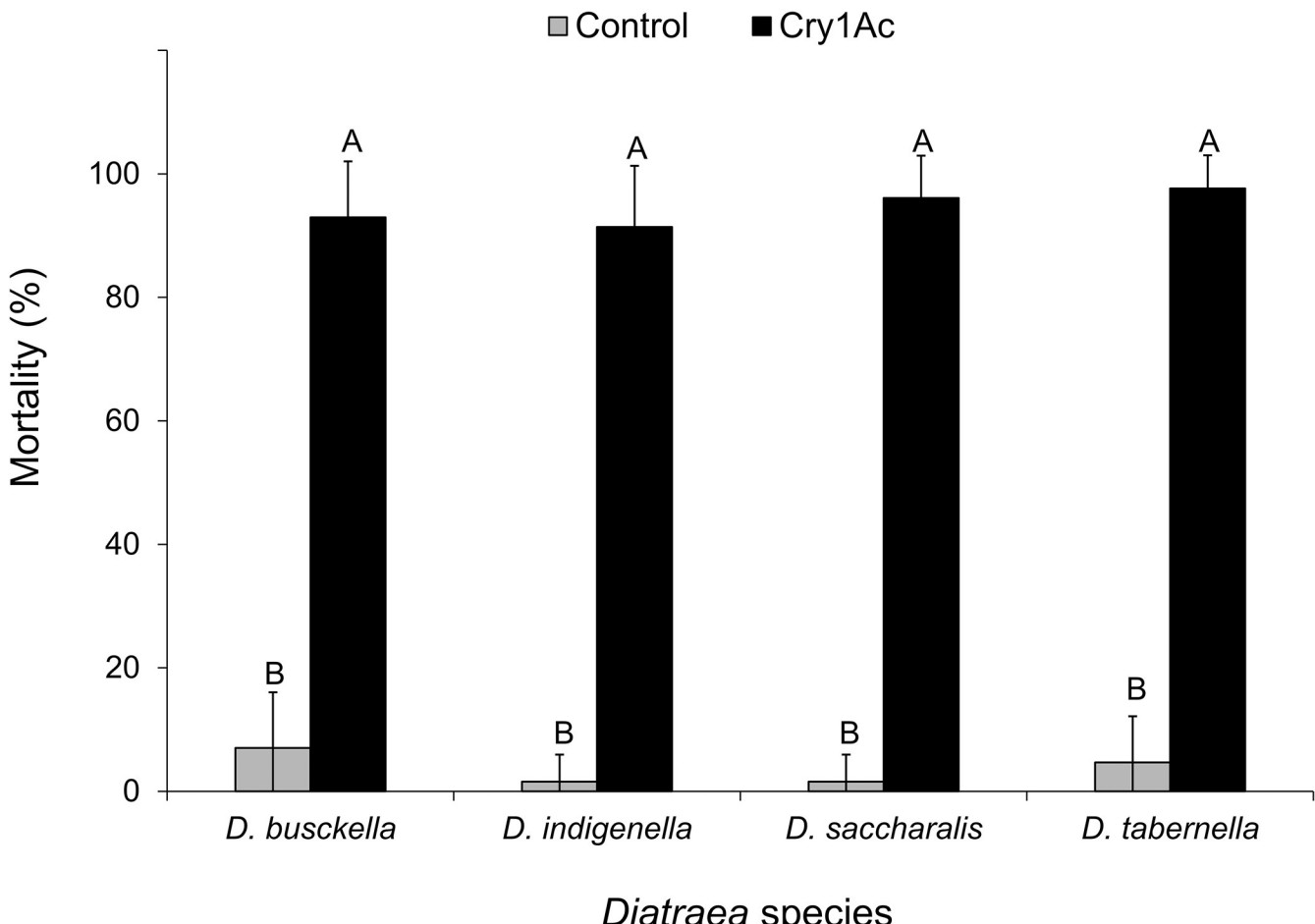

**Fig 1. Mean percent mortality for a diagnostic Cry1Ac dose (24.1 µg/cm$^2$) coated on corn discs against *Diatraea* stem borer species: *D. busckella*, *D. indigenella*, *D. saccharalis*, and *D. tabernella*.** Analyses detected no effect attributable to borer species, but statistically different treatments are separated with uppercase letters within each species (Tukey, α = 0.05).

## Results

### Mortality

Percentage mortality attributable to Cry1Ac (24.1 µg/cm$^2$) consumption was higher than 90% for all the four evaluated species, while control mortality was less than 8% (Fig 1). Analyses detected no effect attributable to borer species (F = 2.18; df = 3, 56; P = 0.099) nor the interaction between borer species and treatment (F = 2.30; df = 3, 56; P = 0.087); however, there was evidence of a treatment effect, with mortality being higher in the Cry1Ac treatment (F = 319.21; df = 1, 56; P < 0.001).

### Growth Index for Survival (GIS) and Relative Growth Index (RGI)

Growth index for survival (GIS) analysis detected an effect attributable to borer species, with *D. indigenella* presenting the highest growth (F = 4.84; df = 3, 56; P = 0.004). A treatment effect was also evidenced, with the GIS being higher in control (F = 3,652.74; df = 1, 56; P < 0.001), as well as an interaction between borer species and treatment, with the control treatment of *D. indigenella* presenting the highest GIS (F = 3.85; df = 3, 56; P = 0.014) (Fig 2). Many larvae reached the second and third instar in the control treatment, while in the Cry1Ac protein

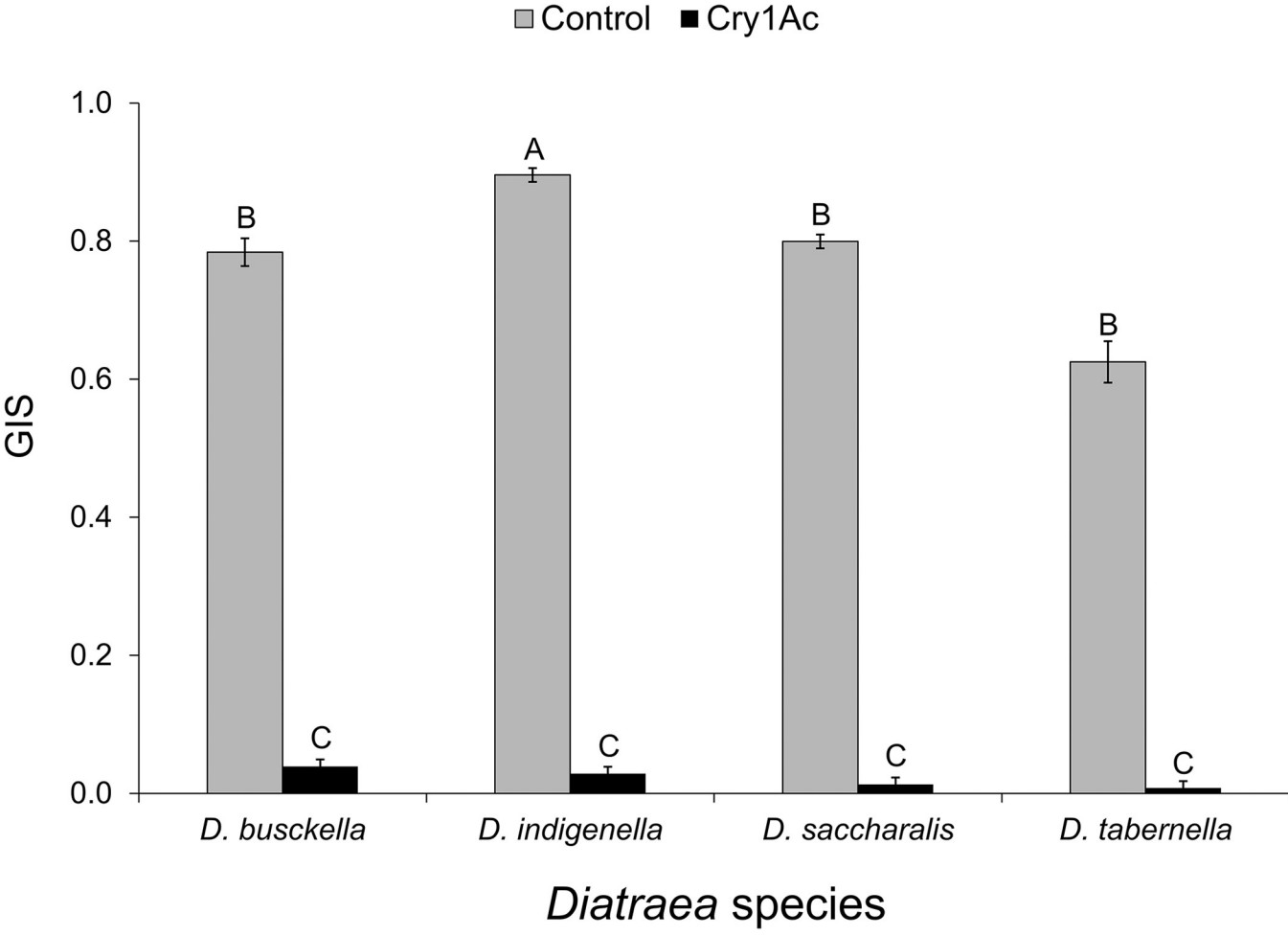

**Fig 2. Mean Growth index for Survival (GIS ± SEM) considering the developmental stage of live individuals (between L1 and L3) of *Diatraea* larvae surviving 7 days after treatment with corn discs coated with Cry1Ac (24.1 µg/cm², compared with control treatment.** Statistically different means for all pairwise comparisons among species and treatments are separated with uppercase letters (Tukey, α = 0.05).

treatment, almost all larvae showed growth inhibition and remained in the first instar (Fig 3). Among *D. busckella* larvae treated with the protein, only six reached the second instar, while control treatments exhibited 89.3% and 90.5% more larvae in the second and third instar, respectively. Similarly, in the same species, 95% more larvae progressed to the second and third instar in control treatments compared to those treated with the protein and observed in the first instar. For *D. indigenella*, control treatments showed 68% and 88% more larvae advancing to the second and third instar, respectively compared to those individuals treated with the protein and observed in the first instar. In *D. saccharalis*, control treatments showed between 93% and 91% more larvae reaching second and third instar, respectively in contrast to those treated with the protein and observed in the first instar. Likewise, in *D. tabernella*, control treatments exhibited between 96% and 93% more larvae moving between the second and third instar, respectively, compared to the individuals treated with the protein and observed in the first instar (Fig 3).

Based on GIS estimates for both treated and control larvae, RGI values in larvae treated with Cry1Ac protein showed levels of growth below 5% compared to those from the controls; with mean values of 0.01 (± 0.01 SEM) for *D. tabernella*, 0.02 (± 0.01 SEM) for *D. saccharalis*,

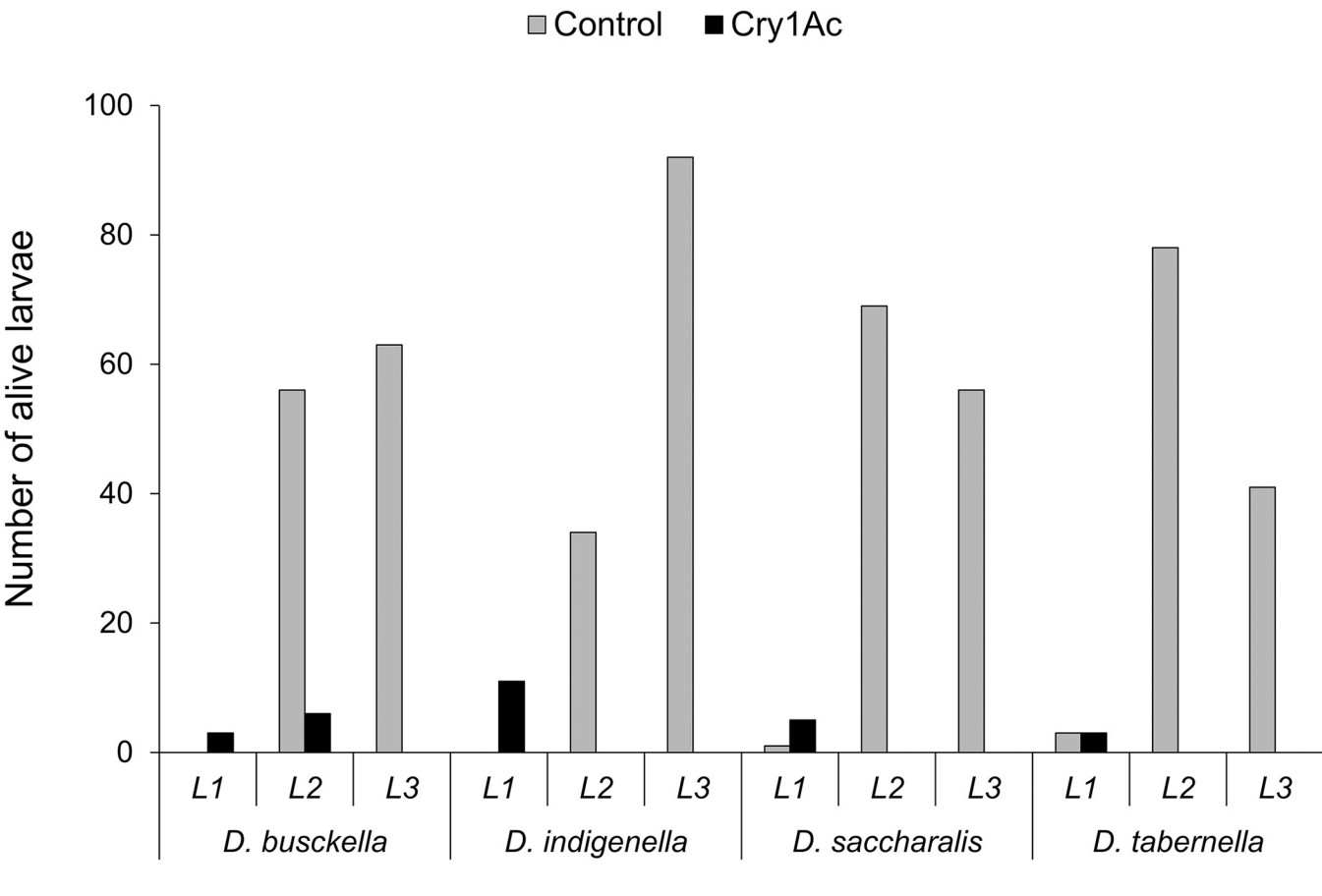

**Fig 3. Number of surviving larvae after feeding on corn discs coated with Cry1Ac (24.1 µg/cm$^2$) and in control.** L1, L2, and L3 refer to the three larval instars observed.

0.03 (± 0.01 SEM) for *D. indigenella*, and 0.05 (± 0.02 SEM) for *D. busckella*; but no significant differences were detected among species (F = 2.04; df = 3, 28; P = 0.134).

### Growth Inhibition for weight (%GI)

The percentage of growth inhibition for the weight (%GI) varied between 93.2% (*D. busckella*), 95.7% (*D. indigenella*), 97.7% (*D. saccharalis*), and 96.3% (*D. tabernella*), with no significant differences between species (F = 2.77; df = 3, 11; P = 0.091). Only 4% (5 larvae) of *D. saccharalis* individuals that consumed Cry1Ac survived and weighted approximately 47-fold lower than the average in control treatment (Fig 4). These larvae did not survive more than one day after the 7-day exposure to treatment when they were then moved to a control diet. Lower weight inhibition was observed in surviving individuals of *D. tabernella* (about 28-fold) and *D. indigenella* (approximately 24-fold) that consumed Cry1Ac, compared to controls (Fig 4), with a survival of only 2% (11 larvae) and 8% (3 larvae), respectively. However, these larvae did not survive more than one day (*D. tabernella*) or survived up to four days (*D. indigenella*) after the 7-days exposure to treatment, when they were then moved to a control diet; in addition, larvae did not show growth or molting during this period. The weight of surviving *D. busckella* individuals was the least affected, with a 16-fold lower weight than in controls (Fig 4), yet the

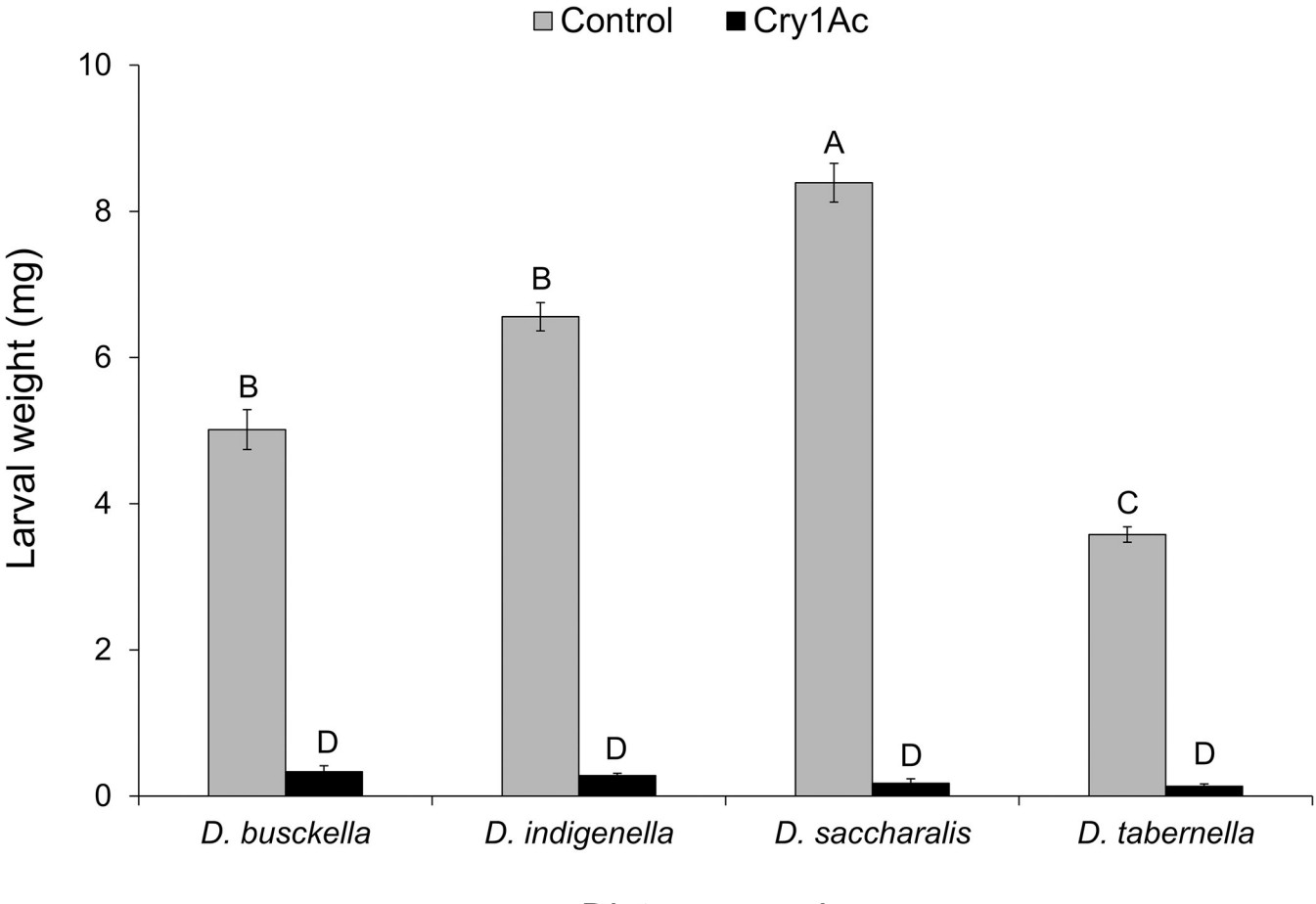

**Fig 4. Mean larval weight (± SEM) of *Diatraea* larvae surviving 7 days after treatment with corn discs coated with Cry1Ac (24.1 μg/cm$^2$), compared with control treatment.** Statistically different means for all pairwise comparisons among species and treatments are separated with uppercase letters (Tukey, α = 0.05).

survival of these larvae was only 7% (9 larvae) and they did not survive more than two days after the 7-days exposure to treatment when they were then moved to a control diet. A general weighted mean among the species showed that larvae survived 1.2 days after the 7-days exposure to treatment.

Analysis of mean larval weights detected an effect of borer species, with *D. saccharalis* being the species that presented the highest weight increase over time and *D. tabernella* the lowest (F = 32.68; df = 3, 56; P < 0.001). Results also showed a treatment effect, with the control treatment presenting a higher average weight than the Cry1Ac treatment (F = 966.69; df = 1, 56; P < 0.001). The interaction between borer species and treatment was also evident, with *D. saccharalis* presenting the highest average weight (F = 32.13; df = 3, 56; P < 0.001) in the control treatment concerning the other species and treatments. In contrast, no differences were found in the weight of surviving larvae on the Cry1Ac treatment (Fig 4).

### Bioassay contamination

The contamination analysis in the corn discs showed an effect of the borer species, with the lowest percentage of contaminated wells occurring in *D. indigenella* (F = 4.40; df = 3, 56,

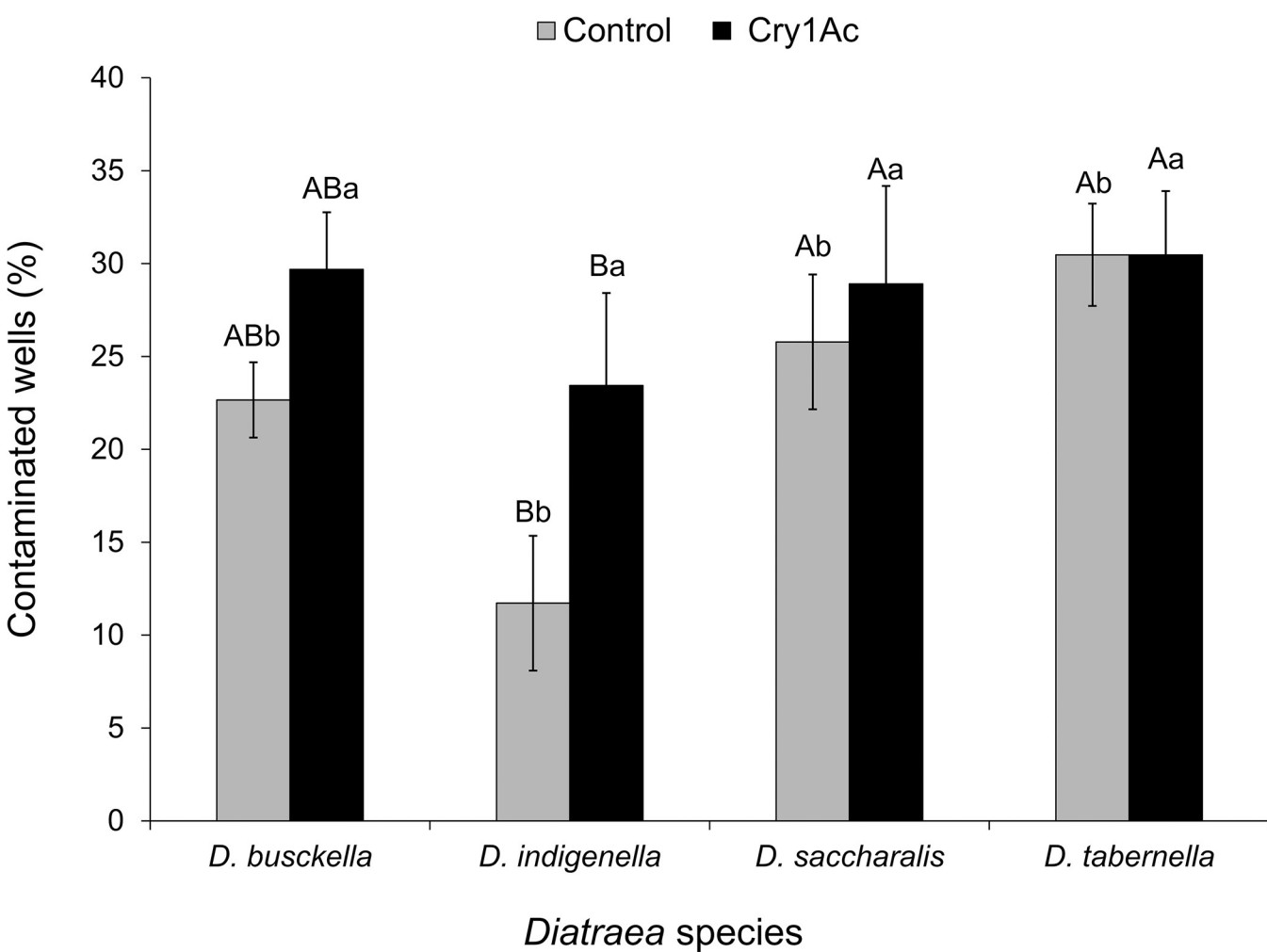

**Fig 5. Percentage of contaminated wells (± SEM) 7 days after coating corn discs with Cry1Ac (24.1 μg/cm$^2$), compared with control treatment with no protein.** Statistically different species are separated with upper case letters within each treatment, while statistically different treatments are separated with lowercase letters within each species (Tukey, α = 0.05).

P = 0.007). A treatment effect was also found, with the percentage of contaminated wells being higher in the Cry1Ac treatment (F = 5.22; df = 1, 56; P = 0.026) and no evidence of interaction between borer species and treatment was detected (F = 1.31; df = 3, 56; P = 0.279) (Fig 5). However, no correlation was found between the percentage of contaminated wells and the percentage of mortality when all species were combined, neither in the control (F = 0.14; df = 1, 30; P = 0.706) nor in the Cry1Ac treatment (F = 1.01; df = 1, 30; P = 0.323).

## Discussion

Bioassay experiments are critical to identifying active insecticidal proteins against *Diatraea* spp., a genus including the most widely distributed boring pests of sugarcane in the western hemisphere [7]. Previous reports presented data evaluating insecticidal proteins from *B. thuringiensis* against *D. saccharalis* [15, 16, 23, 28, 29, 30] and Bt isolates against *D. flavipennella* [21] using artificial diets. The lack of available artificial diets supporting larvae growth has prevented testing susceptibility to these insecticidal proteins in other relevant *Diatraea* species. In resolving this knowledge gap, we developed a new method using corn leaf discs coated with

treatments. We tested its utility in determining susceptibility to the Cry1Ac insecticidal protein in the most widely distributed sugarcane borer species in Colombia. Mortality in control treatment in the developed bioassay protocol did not exceed 10%, as recommended by Beegle [18] when working with artificial diets. Interestingly, we detected weight and developmental differences between the tested species in the control treatment, with *D. saccharalis* presenting the highest average weight and *D indigenella* with the highest number of third instar larvae. This variation is also observed under rearing conditions [26], showing that the bioassay can sustain normal biological function. In contrast to controls, high mortality was observed in Cry1Ac treatments with larvae of *D. saccharalis*, as expected from the increased susceptibility to that protein reported from bioassays using artificial diets for this species [21, 28, 30].

The new bioassay method allowed, for the first time, determining that high susceptibility to Cry1Ac in *D. saccharalis* extends to other relevant sugarcane borer species in Colombia: *D. busckella*, *D. indigenella*, and *D. tabernella*. These observations support the use of this bioassay for testing insecticidal protein activity in *Diatraea* species lacking an artificial diet. However, it is essential to note that if we compare this new tissue bioassay with the one using an artificial diet [21], where a concentration of 619.5 μg/ml would be applied over an area of 1.77 cm$^2$, the use of 30 μl in each well will represent 10.5 μg/cm$^2$ of Cry1Ac, meaning that more than double of the protein is required for coating the tissue (i.e., 24.1 μg/cm$^2$) and effectively expose the insects. Given the challenges and expenses associated with producing Cry1Ac, ensuring a streamlined and cost-effective screening process would necessitate dedicated efforts towards formulating an artificial diet that sufficiently supports the development of different *Diatraea* species.

Bioassays with fresh tissue were previously reported as successful set-ups for testing the activity of Bt insecticidal proteins. Examples include Cry3 and Cry7Aa activity in Coleoptera using potato tubers and potato flour as substrate [35]. However, plant derivatives, such as carotenoids, have been shown to influence Cry1Ac toxicity in bioassays [36]. Using fresh tissue offers the possibility of fungal contamination, as we detected in our method. Treatment with Cry1Ac and subsequent larval death could increase the likelihood of observing fungal contamination, yet we did not detect a correlation between treatment, mortality, and fungal contamination. Contamination was significantly influenced by species, suggesting that some *Diatraea* species may be more prone to harbor/distribute fungal contaminants.

Information on growth inhibition complements mortality data, allowing the detection of possible sublethal effects of Cry proteins. Individuals from all tested species surviving Cry1Ac exposure presented severe growth inhibition, reflected in a significant decrease in weight and stunting as indicated by RGI values, wherein the best of cases, larvae grew only 5% of the expected size considering size in controls. Similar weight reduction was previously reported for *D. saccharalis* exposed to Cry1Ab [23] and Vip3Aa20 [15]. Observations with larvae of *S. frugiperda* [29] and *Ostrinia nubilalis* (Hübner) (Lepidoptera: Crambidae) [28] suggested that heavily stunted larvae (not growing beyond first instar and weighing less than 0.1 mg) after exposure to Bt insecticidal proteins should be considered as dead because they eventually die without completing development. Our observations support this idea, as all surviving larvae presented stunting after Cry1Ac treatment for all species and died shortly after concluding the bioassay.

## Conclusions

This study proposes an alternative, novel method to evaluate Bt insecticidal proteins using fresh corn tissue to determine their toxicity on *Diatraea* species. The bioassay is based on the technical criteria recommended for protocols using artificial diets, such as maintaining control

treatment mortality below 10%. However, sanitation of the work area and careful operations are fundamental to reducing the potential for contamination of the corn substrate during the bioassays, thus helping to obtain consistent results. The high susceptibility to Cry1Ac detected for all tested species, reflected as mortality and heavy stunting of survivors, confirms the potential of Cry proteins for *Diatraea* spp. control. Future work is focused on performing dose-response bioassays and determining toxicity parameters ($LD_{50}$ and $LD_{90}$) for the different *Diatraea* species. These results can serve as a starting point for genetically engineering sugarcane varieties with highly active insecticidal proteins to prevent stem borer attacks. However, considering the challenges and expenses involved in Cry1Ac production, establishing an efficient and economical screening procedure would require focused endeavors in formulating an artificial diet that adequately nurtures the growth of various *Diatraea* species. Future developments in the area would expedite alternatives to the use of chemical insecticides in the control of the stem borers, and it would also need to consider its integration with sugarcane stem borers IPM programs, where biological control is considered a significant pest management factor in several sugar industries in the Americas.

## Supporting information

**S1 Data.**
(XLSX)

**S2 Data.**
(XLSX)

## Acknowledgments

The authors thank Sandra Valencia from the International Center for Tropical Agriculture (CIAT), Orlando Rojas and Alvaro Urresti from the Colombian Sugarcane Research Center (Cenicaña) for providing the logistical and technical support required to conduct this study. We also thank staff from the Insect Molecular Pathology and Resistance laboratory from the Department of Entomology and Plant Pathology of the University of Tennessee, for their work in the insecticidal protein production.

## Author Contributions

**Conceptualization:** Juan Sebastián Ángel-Salazar, Claudia Echeverri-Rubiano, Jairo Rodríguez-Chalarca, Jershon López-Gerena, Juan Luis Jurat-Fuentes, Germán Vargas.

**Data curation:** Juan Sebastián Ángel-Salazar, Claudia Echeverri-Rubiano, Germán Vargas.

**Formal analysis:** Juan Sebastián Ángel-Salazar, Claudia Echeverri-Rubiano, Germán Vargas.

**Funding acquisition:** Jershon López-Gerena, Germán Vargas.

**Investigation:** Juan Sebastián Ángel-Salazar, Claudia Echeverri-Rubiano, Jairo Rodríguez-Chalarca, Jershon López-Gerena, Rafael Ferreira dos Santos, Germán Vargas.

**Methodology:** Juan Sebastián Ángel-Salazar, Claudia Echeverri-Rubiano, Jairo Rodríguez-Chalarca, Jershon López-Gerena, Germán Vargas.

**Project administration:** Jershon López-Gerena, Germán Vargas.

**Resources:** Jairo Rodríguez-Chalarca, Jershon López-Gerena, Rafael Ferreira dos Santos, Juan Luis Jurat-Fuentes, Germán Vargas.

**Software:** Germán Vargas.

**Supervision:** Jershon López-Gerena, Juan Luis Jurat-Fuentes, Germán Vargas.

**Validation:** Claudia Echeverri-Rubiano, Jershon López-Gerena, Rafael Ferreira dos Santos, Alexandra M. Revynthi, Germán Vargas.

**Visualization:** Juan Luis Jurat-Fuentes, Alexandra M. Revynthi, Germán Vargas.

**Writing – original draft:** Juan Sebastián Ángel-Salazar.

**Writing – review & editing:** Juan Sebastián Ángel-Salazar, Claudia Echeverri-Rubiano, Jairo Rodríguez-Chalarca, Jershon López-Gerena, Rafael Ferreira dos Santos, Juan Luis Jurat-Fuentes, Alexandra M. Revynthi, Germán Vargas.

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
