## [Decision Letter · Decision Letter 0]

31 Jul 2023

PONE-D-23-19646Development of a Bioassay Method to Test Activity of Cry Insecticidal Proteins Against Diatraea spp. (Lepidoptera: Crambidae) Sugarcane Stem BorersPLOS ONE

Dear Dr. Vargas,

Thank you for submitting your manuscript to PLOS ONE. After careful consideration, we feel that it has merit but does not fully meet PLOS ONE’s publication criteria as it currently stands. Therefore, we invite you to submit a revised version of the manuscript that addresses the points raised during the review process.

We look forward to receiving your revised manuscript.

Kind regards,

Omaththage P. Perera, Ph.D., FRES

Academic Editor

PLOS ONE

Journal Requirements:

"The  Alliance Bioversity-Ciat (The International Center for Tropical Agriculture) and the Colombian Sugarcane Research Center (Cenicaña) provided logistical and financial support to the first author. The NC246 Multistate Hatch project from the USDA National Institute for Food and Agriculture (NIFA) and the University of Tennessee provided support for insecticidal protein production."

"The authors thank Alliance Bioversity-Ciat (The International Center for Tropical Agriculture) for providing the logistical support required to conduct this study and the Colombian Sugarcane Research Center (Cenicaña) for technical and financial support to the first author. The NC246 Multistate Hatch project from the USDA National Institute for Food and Agriculture (NIFA) and the University of Tennessee provided support for insecticidal protein production."

"The  Alliance Bioversity-Ciat (The International Center for Tropical Agriculture) and the Colombian Sugarcane Research Center (Cenicaña) provided logistical and financial support to the first author. The NC246 Multistate Hatch project from the USDA National Institute for Food and Agriculture (NIFA) and the University of Tennessee provided support for insecticidal protein production."

5. Please remove your figures from within your manuscript file, leaving only the individual TIFF/EPS image files, uploaded separately. These will be automatically included in the reviewers’ PDF.

Additional Editor Comments:

Thank you for your patience during the review process. Three recommendations, two minor and one major revision, were received. Please provide detailed point-by-point responses to all reviewer comments.

Reviewers' comments:

Reviewer's Responses to Questions

**Comments to the Author**

1. Is the manuscript technically sound, and do the data support the conclusions?

Reviewer #1: Yes

Reviewer #2: Yes

Reviewer #3: Partly

2. Has the statistical analysis been performed appropriately and rigorously? 

Reviewer #1: Yes

Reviewer #2: Yes

Reviewer #3: Yes

3. Have the authors made all data underlying the findings in their manuscript fully available?

Reviewer #1: Yes

Reviewer #2: Yes

Reviewer #3: No

4. Is the manuscript presented in an intelligible fashion and written in standard English?

Reviewer #1: Yes

Reviewer #2: Yes

Reviewer #3: Yes

5. Review Comments to the Author

Reviewer #1: This is an interesting adaptation of a protocol to assess the susceptibility of sugarcane borers. If thoroughly explained, it can become a standard protocol for others to follow. Some of the necessary additions and clarifications are:

Line 103: ‘Corn cobs’ requires a full description of the maize cultivar, its agronomic management, the test to verify that the cultivar did not express any Bt proteins, when it was harvested and how, etc.

Lines 135-137: …24.1 μg/cm2 of Cry1Ac…was selected in assessing toxicity based on previously reported LC50 values for Cry1Ac from diet surface contamination bioassays with D. saccharalis and D. flavipennella. According to Lemes et al. [20], the LC50 for the latter is LC50 = 495ng∕cm2 Why the difference?

Lines 275-277: The lack of available artificial diets supporting larvae growth has prevented testing susceptibility to these insecticidal proteins in other relevant Diatraea species. According to the text before and after this statement, THERE IS artificial diet for D. saccharalis, and D. flavipennella. Perhaps a little bit of effort with successful artificial diets for these two pests can be adapted for the other two. To propose your protocol as a ‘standard’, a full description of the maize cultivar, its growth and other parameters need to be provided. Has this protocol tested with other cultivars and on different leaves?

Lines 297-298: …meaning that more than double of the protein is required for coating the tissue (i.e., 24.1 μg/cm2) and effectively expose the insects... I would be important to note that production of Cry1Ac is not easy, neither cheap. Of greater value could be the cost comparison between this method and the use of artificial diet.

In the conclusions section I could not find anything different of what was described in the abstract or complemented in the text. Is it necessary?

Minor comments.

Line 11: I believe it should be CIAT

Line 30: Because there is no described artificial diet and availability of most purified Bt toxins,…

Line 39: Remove species evaluated

Line 47 mentions: cause significant yield losses. Provide the range, rather than requiring the reader to search for references 3 and 4.

Lines 51 and 52: with production losses (Provide the range) depending on the crop stage and damage caused (Describe)

Line 56: infection by Coletotrichum falcatum, a fungus that resultsing

Line 58: …as a reduction in both total biomass production and sucrose content. Provide ranges.

Line 75: application Suggestions; alternative / option

Line 78: , or by direct observation of their efficacy in plants in the field (Dively et al. 2023 [https://www.mdpi.com/2075-4450/14/7/577], and previous publications)

Line 97: Entomology Laboratory does not require capital letters. It is only ‘a room’.

Line 103: Fix F1

Lines 104-105: if the thickness of the cob has a variation of 20%, then the term ‘uniform’ disk is not correct. Just saying.

Line 126: on four Diatraea species, Diatraea. saccharalis

Lines 127-128: By the time maize is ready to provide corn discs 75 days after planting, a plant may have at least 12 leaves at different stages of development and thickness. Provide information on the precise development stage of maize using the ISU scale (https://store.extension.iastate.edu/product/Corn-Growth-and-Development), and describe which maize leaf(ves) was used in the bioassays.

Line 133: was …water and detergent alone… only used in the control? If so, specify / justify why.

Line 141: Describe how Mmortality was scored 7 days after infestation

Lines 153-155: Is it necessary to describe this simple formula? The same one described in line 188.

Line 196: It is described that “The 16 wells within each of the eight replicates were considered subsamples” Is this a replication, made only once with one corn borer generation? Specify.

Line 225: “almost all larvae were stunted” requires precision.

Reviewer #2: The Ms. has shown interesting work however the following comments need to address in order to improve the understanding of the research.

1. In Introduction, Authors need to incorporate one paragraph about the previous diet formulations studied and what were the drawback of those systems.

2. Authors need to check the typographical errors i.e. Line no. 156 Metter Toledo AB204 must be Mettler Toledo AB204

3. Authors need to explain the except fresh corn tissue used in the present study whether it was leaf or cob or any other fresh part. Also explain the age of the fresh tissue, whether it can stored or not so that assay need to perform throughout the year.

4. In order to get consistent results, authors need to provide some proximate nutritional analysis of the fresh tissue used for bioassay.

5. Authors need to explain how the protein is absorbed and spreaded uniform to fresh tissue.

Reviewer #3: PONE-D-23-19646

“Development of a Bioassay Method to Test Activity of Cry Insecticidal Proteins Against

Diatraea spp. (Lepidoptera: Crambidae) Sugarcane Stem Borers” by Ángel-Salazar et al.

This paper describes a Bacillus thuringiensis (Bt) insecticidal protein bioassay method for sugarcane borers based on Bt overlay of corn leaf discs. This assay was stated as necessary due to lack of an artificial diet for D. saccharalis. Assays are restricted to use of a “diagnostic dose” determined from prior experiments on other Diatraea species.

Overall, the paper describes developmental as opposed to experimental work. Specifically the work done covers documentation of larval performance at a single insecticidal protein dose and other factors influencing assay reproducibility (e.g. fungal contamination). Experimental components such as determination of LC50 or LC99 concentrations in a dose-response bioassay experiments were not included, although this was stated in the Conclusions as likely future work. Thus, this developmental work might be better suited for a methods development type journal.

Additionally, assays were negatively influenced by fungal contamination which authors state as a significantly impacting results between species in the authors comparisons. This significant variation in contamination likely influences larval survival or performance, and should be addressed prior to publication of these methods. Surface sterilization of leaf discs by dipping in a dilute bleach solution (~2-3%) may lessen the amount of observed contamination in these assays.

Data for survivorship post-movement to non-Bt diet should be provided (e.g. proportional survivorship)

Specific comments:

Line 100 Please define "CRV":

Line 128 Please indicate: 1. What hybrid was used, 2. field or greenhouse grown 2. V-stage range of plants; can vary at d after planting depending on hybrid maturity date and growth conditions.

Line 137 “diet surface contamination bioassays with D. saccharalis”. Contamination seems out of place.

Line 144 “…wells, was used to set a species with the protein, while another tray had the control”. The portion “was used to set a species with the protein” is unclear.

Line 225: Would suggest using “growth inhibition” instead of “stunted”

Lines 243-246 states “These larvae did not survive more than a day (D. tabernella) or survived

up to four days (D. indigenella) after the 7-days exposure to treatment, when they were

then moved to a control diet; in addition, larvae did not show growth or molting during this

period.”, but this is not described in the Methods. Similarly for line 248.

Lines 261-263 describes a significant effect of contamination. Granted, one would suspect that the Cry1Ac protien in this material would have a greater effect on mortality, but could this difference mortality compared to control also be influenced by corresponding differences in fungal contamination? Could this difference in fungal contamination also contribute to a portion of the differences in other metrics? There should be a way to tease out contribution of these two by determining effects of fungal contamination alone. A suggestion would be to surface sterilize plant leaves using a dilute bleach solution (2-3%) to reduce contamination rates.

Line 286 What is meant by "reproducible mortality"? Low variation between replicates or consistency with prior experiments on artificial diet? The latter can be assumed. Consider changeing "high(ly) reproducible mortality" with "high mortality'.

Line 304-305 “subsequent larval death could increase the likelihood of observing fungal contamination”. Are authors suggesting that fungus may be feeding on the cadavers? Could init be equally likely that the fungus is contributing to larval mortality?

Lines 319-320 “all surviving larvae presented stunting after Cry1Ac treatment for all species and died shortly after concluding the bioassay”. The methods for this need to be described and the data shown at least in a Supplementary table, or provide mean number of days survived post 7 d exposure.

Line 330 Change “Control” to “control”

Line 331 Do authors mean “dose-response bioassays”?

6. PLOS authors have the option to publish the peer review history of their article (what does this mean?). If published, this will include your full peer review and any attached files.

Reviewer #1: **Yes: **Carlos A. Blanco

Reviewer #2: **Yes: **Devendra Jain

Reviewer #3: No

---

## [Author Response · Author response to Decision Letter 0]

13 Sep 2023

Reviewer #1: 

This is an interesting adaptation of a protocol to assess the susceptibility of sugarcane borers. If thoroughly explained, it can become a standard protocol for others to follow. Some of the necessary additions and clarifications are:

Line 103: ‘Corn cobs’ requires a full description of the maize cultivar, its agronomic management, the test to verify that the cultivar did not express any Bt proteins, when it was harvested and how, etc.

R/ In Lines 119-127 we have indicated “The maize cultivar SV 1035 was grown conventionally (mineral fertilization with NPK and manual weed control), but without using insecticides, while ensuring it was situated 2 kilometers away from other commercial corn crops. This strategic separation aimed to prevent any potential cross-contamination. The harvesting process was done manually at 70 days of the crop cycle. Coincidentally, this same source of corn was also employed as a substrate to sustain the various Diatraea species stock colonies at the Cenicaña entomology laboratory. To confirm the absence of any expression of Bt proteins, we conducted regular assessments of survival within the stock colonies.” 

Lines 135-137: …24.1 μg/cm2 of Cry1Ac…was selected in assessing toxicity based on previously reported LC50 values for Cry1Ac from diet surface contamination bioassays with D. saccharalis and D. flavipennella. According to Lemes et al. [20], the LC50 for the latter is LC50 = 495ng∕cm2 Why the difference?

R/ Lemes et al. [21] reported that Cry1Ac exhibited the highest toxicity towards D. flavipennella and E. lignosellus, with LC50 values of 8.6 and 15.6 ng/cm2, respectively. Based on this information, we anticipated that employing an application rate greater than that used on D. flavipennella would result in increased mortality in our different Diatraea species in this new proposed protocol. The latter was also developed on several trial-and-error observations trying to adjust an efficacious rate on this fresh tissue.

Lines 275-277: The lack of available artificial diets supporting larvae growth has prevented testing susceptibility to these insecticidal proteins in other relevant Diatraea species. According to the text before and after this statement, THERE IS artificial diet for D. saccharalis, and D. flavipennella. Perhaps a little bit of effort with successful artificial diets for these two pests can be adapted for the other two. To propose your protocol as a ‘standard’, a full description of the maize cultivar, its growth and other parameters need to be provided. Has this protocol tested with other cultivars and on different leaves?

R/ We acknowledge the reviewer’s observation regarding the ‘relative’ effort required to develop an artificial diet for the additional three Diatraea species, namely busckella, indigenella and tabernella. We believe that investing additional effort will yield appropriate diets for these species. However, a potential challenge arises from the necessity of utilizing distinct diets for each species, which could complicate the comparability of our trials. This complexity implies that even if we were to have access to these diets, evaluating and contrasting the survival rates of different species fed on disparate diets might hinder the straightforward comparisons achieved in our analysis.

In presenting our protocol as a new ‘standard’, we are providing a full description of the maize cultivar’s management (Lines 119-124). Nonetheless, we must acknowledge that while we propose this protocol as a robust framework, its applicability has not been tested with alternate cultivars or utilizing different plant tissues than those delineated in our description.

Lines 297-298: …meaning that more than double of the protein is required for coating the tissue (i.e., 24.1 μg/cm2) and effectively expose the insects... I would be important to note that production of Cry1Ac is not easy, neither cheap. Of greater value could be the cost comparison between this method and the use of artificial diet.

R/ We acknowledge the reviewer’s consideration of the costs associated with our proposed protocol. Beyond underscoring the significance of acquiring knowledge concerning the varied response of Diatraea species to Cry proteins before embarking on any transformation endeavors, we have indicated in Lines 377-380, that “Given the challenges and expenses associated with producing Cry1Ac, ensuring a streamlined and cost-effective screening process would necessitate dedicated efforts towards formulating an artificial diet that sufficiently supports the development of different Diatraea species.” 

In the conclusions section I could not find anything different of what was described in the abstract or complemented in the text. Is it necessary?

R/ As per your comments on the cost aspects of our protocol, an extra statement has been added in Lines 416-419, indicating: “… However, considering the challenges and expenses involved in Cry1Ac production, establishing an efficient and economical screening procedure would require focused endeavors in formulating an artificial diet that adequately nurtures the growth of various Diatraea species.” 

Minor comments.

Line 11: I believe it should be CIAT

R/ Done

Line 30: Because there is no described artificial diet and availability of most purified Bt toxins,…

R/ Text has modified in lines 31-33: “Because there is no described artificial diet for borer species other than D. saccharalis and availability of most purified Bt toxins is restricted, this study aimed at …”

Line 39: Remove species evaluated

R/ We removed the word ‘evaluated’

Line 47 mentions: cause significant yield losses. Provide the range, rather than requiring the reader to search for references 3 and 4.

R/ A range of losses has been indicated in Lines 48-49: “… ranging from 100 to 140 kilograms of sucrose per hectare, and per each percent of bored internodes.”

Lines 51 and 52: with production losses (Provide the range) depending on the crop stage and damage caused (Describe)

R/ In the following lines (from Line 54) we describe the damage and now we have incorporated the ranges of crop losses. See Lines 58, 60-63.

Line 56: infection by Coletotrichum falcatum, a fungus that resultsing

R/ Done

Line 58: …as a reduction in both total biomass production and sucrose content. Provide ranges.

R/ Done. See Lines 60-63.

Line 75: application Suggestions; alternative / option

R/ Done. Now in Line 80 we included ‘alternative.’

Line 78: , or by direct observation of their efficacy in plants in the field (Dively et al. 2023 [https://www.mdpi.com/2075-4450/14/7/577], and previous publications)

R/ New ref and comment have been included in Lines 83-84. 

Line 97: Entomology Laboratory does not require capital letters. It is only ‘a room’.

R/ Done.

Line 103: Fix F1

R/ In the text we have changed the term to: “first generation individuals …” (Line 116)

Lines 104-105: if the thickness of the cob has a variation of 20%, then the term ‘uniform’ disk is not correct. Just saying.

R/ We have excluded the term ‘uniform’

Line 126: on four Diatraea species, Diatraea. Saccharalis

R/ The genus and species descriptor are now included in Line 86

Lines 127-128: By the time maize is ready to provide corn discs 75 days after planting, a plant may have at least 12 leaves at different stages of development and thickness. Provide information on the precise development stage of maize using the ISU scale (https://store.extension.iastate.edu/product/Corn-Growth-and-Development), and describe which maize leaf(ves) was used in the bioassays.

R/ Indication has been made in Line 149-150 that we used cob corn discs, when kernels were at the milk stage (R3). No corn leaves were used in our trials.

Line 133: was …water and detergent alone… only used in the control? If so, specify / justify why.

R/ To account for any effects of moisturizing the cob corn discs with water and detergent (Triton-X 100), control larvae were exposed to discs treated with water and detergent. This is now indicated in Lines 159-161.

Line 141: Describe how Mmortality was scored 7 days after infestation

R/ Now in Line 173 it reads: “… after infestation, by assessing any level of activity on treated larvae.”

Lines 153-155: Is it necessary to describe this simple formula? The same one described in line 188.

R/ We consider important to describe these simple formulas, so the reader can easily understand and reproduce the variables described.

Line 196: It is described that “The 16 wells within each of the eight replicates were considered subsamples” Is this a replication, made only once with one corn borer generation? Specify.

R/ Now in Lines 232-233 we specify that “Each combination of Diatraea species and the protein was made performed once using a single borer generation.”

Line 225: “almost all larvae were stunted” requires precision.

R/ An addition on the text has been made on Line 268, indicating that: “Among D. busckella larvae treated with the protein, only six reached the second instar, while control treatments exhibited 89.3% and 90.5% more larvae in the second and third instar, respectively. Similarly, in the same species, 95% more larvae progressed to the second and third instar in control treatments compared to those treated with the protein and observed in the first instar. For D. indigenella, control treatments showed 68% and 88% more larvae advancing to the second and third instar, respectively compared to those individuals treated with the protein and observed in the first instar. In D. saccharalis, control treatments showed between 93% and 91% more larvae reaching second and third instar, respectively in contrast to those treated with the protein and observed in the first instar. Likewise, in D. tabernella, control treatments exhibited between 96% and 93% more larvae moving between the second and third instar, respectively, compared to the individuals treated with the protein and observed in the first instar (Fig 3).” 

Reviewer #2: 

The Ms. has shown interesting work however the following comments need to address in order to improve the understanding of the research.

1. In Introduction, Authors need to incorporate one paragraph about the previous diet formulations studied and what were the drawback of those systems.

R/We have dedicated an additional paragraph on relation to the issue of the available diets. Now in Lines 93-100 it reads like this: “Additional research is warranted in relation to the development of an artificial diet for the additional Diatraea species (i.e., D. busckella, D. indigenella and D. tabernella). Investing additional effort will yield appropriate diets for these species. However, a potential challenge arises from the necessity of utilizing distinct diets for each species, which could complicate the comparability of survival and biological responses in trials where comparisons among different species are required. Performing experiments using a species-specific diet, would increase the variation in the set-up, hindering this was straightforward comparisons among Diatraea species.

2. Authors need to check the typographical errors i.e. Line no. 156 Metter Toledo AB204 must be Mettler Toledo AB204

R/ Done.

3. Authors need to explain the except fresh corn tissue used in the present study whether it was leaf or cob or any other fresh part. Also explain the age of the fresh tissue, whether it can stored or not so that assay need to perform throughout the year.

R/ In line 149 it is now indicated that we used cob corn discs 70–75 days after planting, when kernels were at the Milk stage (R3). Also in Line 154, we indicated that non-peeled cob corns were stored for nor more than 1 week at 8°C. 

4. In order to get consistent results, authors need to provide some proximate nutritional analysis of the fresh tissue used for bioassay.

R/ As we have indicated before we used cob corn discs at the milk stage (R3). In addition, we are now indicating in Lines 151-153 that: “In general, the corn kernel at the milk stage is on the cusp of maturity, yet it remains soft and brimming with a milky fluid. This fluid is rich in carbohydrates, mainly in the form of sugars. At the start of the R3 stage, approximately 80% of its content comprises water.” 

5. Authors need to explain how the protein is absorbed and spreaded uniform to fresh tissue.

R/ In Lines 168-170 it has been indicated that “… the volume of 60 µl was uniformly applied onto the corn tissue using a 100 µl Eppendorf pipette. The tray was then balanced after each application to facilitate protein dispersion and absorption within the tissue.” 

Reviewer #3: PONE-D-23-19646

“Development of a Bioassay Method to Test Activity of Cry Insecticidal Proteins Against

Diatraea spp. (Lepidoptera: Crambidae) Sugarcane Stem Borers” by Ángel-Salazar et al.

This paper describes a Bacillus thuringiensis (Bt) insecticidal protein bioassay method for sugarcane borers based on Bt overlay of corn leaf discs. This assay was stated as necessary due to lack of an artificial diet for D. saccharalis. Assays are restricted to use of a “diagnostic dose” determined from prior experiments on other Diatraea species.

R/ In line 149 we are indicating that we used cob corn discs instead of leaf discs. 

Overall, the paper describes developmental as opposed to experimental work. Specifically the work done covers documentation of larval performance at a single insecticidal protein dose and other factors influencing assay reproducibility (e.g. fungal contamination). Experimental components such as determination of LC50 or LC99 concentrations in a dose-response bioassay experiments were not included, although this was stated in the Conclusions as likely future work. Thus, this developmental work might be better suited for a methods development type journal.

R/ We acknowledge the reviewer’s perspective, yet we maintain that our manuscript focuses primarily on introducing a novel method. This emphasis aligns with the journal’s publication scope for research articles, supported by the criteria of utility (providing an alternative to the challenges of testing proteins on various Diatraea species), validation (demonstrating the toxicity of the purified protein across four different Diatraea species using fresh tissue as a proof-of-principle), and availability, as we offer a comprehensive description of the materials and methods, ensuring reproducibility in any laboratory.

Additionally, assays were negatively influenced by fungal contamination which authors state as a significantly impacting results between species in the authors comparisons. This significant variation in contamination likely influences larval survival or performance, and should be addressed prior to publication of these methods. Surface sterilization of leaf discs by dipping in a dilute bleach solution (~2-3%) may lessen the amount of observed contamination in these assays.

R/ we agree with the reviewer that “sanitation of the work area and careful operations are fundamental to reducing the potential for contamination of the corn substrate during the bioassays, thus helping to obtain consistent results” (Lines 408-410). However, we also indicated that “the bioassay is based on the technical criteria recommended for protocols using artificial diets, such as maintaining control treatment mortality below 10%” (Lines 406-408). Therefore, contamination was not significantly impacting survival as we “did not detect a correlation between treatment, mortality, and fungal contamination” (Lines 387-388).

Data for survivorship post-movement to non-Bt diet should be provided (e.g. proportional survivorship)

R/ Percent survival after the seven days exposure to the protein is now indicated for D. saccharalis in Line 301, for D. tabernella and D. indigenella in Line 307, respectively; and for D. busckella in Line 313. Is it noteworthy that no larvae survived longer than 4 days after the exposure to the protein and the post-movement to a non-Bt diet.

Specific comments:

Line 100 Please define "CRV":

R/ CRV is defined in Line 69

Line 128 Please indicate: 1. What hybrid was used, 2. field or greenhouse grown 2. V-stage range of plants; can vary at d after planting depending on hybrid maturity date and growth conditions.

R/ In Line 119 we are indicating that we used the hybrid SV 1035, under field conditions (Line 121), while harvest of the corn cobs was done when kernels were at the Milk stage (Line 150). 

Line 137 “diet surface contamination bioassays with D. saccharalis”. Contamination seems out of place.

R/ the term ‘contamination’ has been eliminated in Line 165.

Line 144 “…wells, was used to set a species with the protein, while another tray had the control”. The portion “was used to set a species with the protein” is unclear.

R/ In lines 178-180, the text has been modified for clarity: “Each combination of Diatraea species and the protein were evaluated separately using a tray containing 16 wells. Simultaneously, a separate tray was designated as the control, at it contained water and detergent.”

Line 225: Would suggest using “growth inhibition” instead of “stunted”

R/ In line 267 the term “growth inhibition” replaced “stunted.”

Lines 243-246 states “These larvae did not survive more than a day (D. tabernella) or survived up to four days (D. indigenella) after the 7-days exposure to treatment, when they were then moved to a control diet; in addition, larvae did not show growth or molting during this period.”, but this is not described in the Methods. Similarly for line 248.

R/ In Materials and methods (Lines 174-176) it has been indicated that: “Larvae that survived the protein treatment were subsequently transferred to a control diet (corn cob discs treated with water and detergent) to observe further development.”

Lines 261-263 describes a significant effect of contamination. Granted, one would suspect that the Cry1Ac protien in this material would have a greater effect on mortality, but could this difference mortality compared to control also be influenced by corresponding differences in fungal contamination? Could this difference in fungal contamination also contribute to a portion of the differences in other metrics? There should be a way to tease out contribution of these two by determining effects of fungal contamination alone. A suggestion would be to surface sterilize plant leaves using a dilute bleach solution (2-3%) to reduce contamination rates.

R/ we acknowledge the reviewer’s concern regarding contamination. In our discussion, we indicate that “… treatment with Cry1Ac and subsequent larval death could increase the likelihood of observing fungal contamination …” (Lines 386-387), while we considered that the effect of contamination on mortality was not significant as no correlation was found between mortality and contamination. The latter is evident in Fig 5, where fungal contamination reached as high as 30% in both treated and non-treated D. tabernella larvae, resulting in mortalities of 4.7% and 97.7% for control larvae and treated larvae, respectively. While we appreciate the suggestion of using a diluted bleach solution, it is important to note that we employed corn cob discs as the corn tissue substrate. Consequently, we opted to avoid introducing any substance that could be absorbed by the tissue and potentially impact larval survival. 

Line 286 What is meant by "reproducible mortality"? Low variation between replicates or consistency with prior experiments on artificial diet? The latter can be assumed. Consider changeing "high(ly) reproducible mortality" with "high mortality'.

R/ the term ‘high reproducible mortality’ was replaced by ‘high mortality’ (Now in Line 366).

Line 304-305 “subsequent larval death could increase the likelihood of observing fungal contamination”. Are authors suggesting that fungus may be feeding on the cadavers? Could init be equally likely that the fungus is contributing to larval mortality?

R/ We addressed the issue of contamination and mortality in a previous comment.

Lines 319-320 “all surviving larvae presented stunting after Cry1Ac treatment for all species and died shortly after concluding the bioassay”. The methods for this need to be described and the data shown at least in a Supplementary table, or provide mean number of days survived post 7 d exposure.

R/ In Lines 173-175 it is now has indicated the way surviving larvae were observed. In addition, a description of the number of individuals surviving after the protein exposure is indicated in Lines 299-311, where additionally a weighted mean number of days survived post 7-day treatment in now indicated in Line 315. 

Line 330 Change “Control” to “control”

R/ Done

Line 331 Do authors mean “dose-response bioassays”?

R/ Now in Line 413, the term ‘dose-response bioassays’ is indicated.

---

## [Decision Letter · Decision Letter 1]

4 Oct 2023

Development of a Bioassay Method to Test Activity of Cry Insecticidal Proteins Against Diatraea spp. (Lepidoptera: Crambidae) Sugarcane Stem Borers

PONE-D-23-19646R1

Dear Dr. Vargas,

We’re pleased to inform you that your manuscript has been judged scientifically suitable for publication and will be formally accepted for publication once it meets all outstanding technical requirements.

Kind regards,

Omaththage P. Perera, Ph.D., FRES

Academic Editor

PLOS ONE

Reviewers' comments:

Reviewer's Responses to Questions

**Comments to the Author**

1. If the authors have adequately addressed your comments raised in a previous round of review and you feel that this manuscript is now acceptable for publication, you may indicate that here to bypass the “Comments to the Author” section, enter your conflict of interest statement in the “Confidential to Editor” section, and submit your "Accept" recommendation.

Reviewer #1: All comments have been addressed

Reviewer #2: All comments have been addressed

2. Is the manuscript technically sound, and do the data support the conclusions?

Reviewer #1: Yes

Reviewer #2: Yes

3. Has the statistical analysis been performed appropriately and rigorously? 

Reviewer #1: Yes

Reviewer #2: Yes

4. Have the authors made all data underlying the findings in their manuscript fully available?

Reviewer #1: Yes

Reviewer #2: Yes

5. Is the manuscript presented in an intelligible fashion and written in standard English?

Reviewer #1: Yes

Reviewer #2: Yes

6. Review Comments to the Author

Reviewer #1: None really, good work. Nothing else to add at the moment but the system requires certain characters to be entered.

Reviewer #2: The Ms. was revisited after addressing all the comment from suggested and improved substantially and may be accepted for publication.

7. PLOS authors have the option to publish the peer review history of their article (what does this mean?). If published, this will include your full peer review and any attached files.

Reviewer #1: **Yes: **Carlos A. Blanco

Reviewer #2: **Yes: **Devendra Jain

---

## [Editor Report · Acceptance letter]

10 Oct 2023

PONE-D-23-19646R1 

Development of a Bioassay Method to Test Activity of Cry Insecticidal Proteins Against *Diatraea* spp. (Lepidoptera: Crambidae) Sugarcane Stem Borers 

Dear Dr. Vargas:

I'm pleased to inform you that your manuscript has been deemed suitable for publication in PLOS ONE. Congratulations! Your manuscript is now with our production department. 

Kind regards, 

on behalf of

Dr. Omaththage P. Perera 

Academic Editor

PLOS ONE